# Aspects of Nanotechnology for COVID-19 Vaccine Development and Its Delivery Applications

**DOI:** 10.3390/pharmaceutics15020451

**Published:** 2023-01-30

**Authors:** Pranav Kumar Prabhakar, Navneet Khurana, Manish Vyas, Vikas Sharma, Gaber El-Saber Batiha, Harpreet Kaur, Jashanpreet Singh, Deepak Kumar, Neha Sharma, Ajeet Kaushik, Raj Kumar

**Affiliations:** 1School of Allied Medical Sciences, Lovely Professional University, Punjab 144411, India; 2School of Pharmaceutical Sciences, Lovely Professional University, Punjab 144411, India; 3Department of Pharmacology and Therapeutics, Faculty of Veterinary Medicine, Damanhour University, Damanhour 22511, Egypt; 4School of Chemical Engineering and Physical Sciences, Lovely Professional University, Punjab 144411, India; 5NanoBioTech Laboratory, Department of Environmental Engineering, Florida Polytechnic University, Lakeland, FL 33805, USA; 6School of Engineering, University of Petroleum and Energy Studies (UPES), Uttarakhand 248007, India; 7Department of Pharmaceutical Sciences, University of Nebraska Medical Sciences, Omaha, NE 68198, USA

**Keywords:** nanotechnology, COVID-19, virus-vectored, mRNA, diagnosis, vaccine, mutation

## Abstract

Coronavirus, a causative agent of the common cold to a much more complicated disease such as “severe acute respiratory syndrome (SARS-CoV-2), Middle East Respiratory Syndrome (MERS-CoV-2), and Coronavirus Disease 2019 (COVID-19)”, is a member of the coronaviridae family and contains a positive-sense single-stranded RNA of 26–32 kilobase pairs. COVID-19 has shown very high mortality and morbidity and imparted a significantly impacted socioeconomic status. There are many variants of SARS-CoV-2 that have originated from the mutation of the genetic material of the original coronavirus. This has raised the demand for efficient treatment/therapy to manage newly emerged SARS-CoV-2 infections successfully. However, different types of vaccines have been developed and administered to patients but need more attention because COVID-19 is not under complete control. In this article, currently developed nanotechnology-based vaccines are explored, such as inactivated virus vaccines, mRNA-based vaccines, DNA-based vaccines, S-protein-based vaccines, virus-vectored vaccines, etc. One of the important aspects of vaccines is their administration inside the host body wherein nanotechnology can play a very crucial role. Currently, more than 26 nanotechnology-based COVID-19 vaccine candidates are in various phases of clinical trials. Nanotechnology is one of the growing fields in drug discovery and drug delivery that can also be used for the tackling of coronavirus. Nanotechnology can be used in various ways to design and develop tools and strategies for detection, diagnosis, and therapeutic and vaccine development to protect against COVID-19. The design of instruments for speedy, precise, and sensitive diagnosis, the fabrication of potent sanitizers, the delivery of extracellular antigenic components or mRNA-based vaccines into human tissues, and the administration of antiretroviral medicines into the organism are nanotechnology-based strategies for COVID-19 management. Herein, we discuss the application of nanotechnology in COVID-19 vaccine development and the challenges and opportunities in this approach.

## 1. Introduction

Coronavirus disease 2019, commonly known as COVID-19, is caused by a new type of coronavirus that belongs to the coronaviridae family of viruses. This virus family is known to cause the simplest common cold and some more complicated diseases such as “severe acute respiratory syndrome (SARS-CoV)” and “Middle East Respiratory Syndrome (MERS-CoV)” [1]. It is an enveloped virus whose genetic material contains a single-stranded positive (+ve)-sense polyadenylated non-segmented ribose nucleic acid (RNA) of 25–33 kilobase pairs [2]. The name coronavirus is because its structure has a crown-like appearance due to proteinous spikes. There are four types of coronaviruses that have been discovered named alpha (α), beta (β), gamma (γ), and delta (δ) [3,4]. Out of these four types, only alpha and beta coronaviruses infect humans. To date, seven different types of human-infecting coronaviruses (HCoVs) have been found, and these are HCoV-229E, HCoV-NL63, HCoV-HKU1, HCoV-OC43, MERS-CoV, SARS-CoV, and SARS-CoV-2. The human-infecting alpha coronaviruses are HCoV-229E and HCoV-NL63, and those of the beta coronavirus type are MERS-CoV, SARS-CoV, HCoV-OC43, HCoV-HKU1, and SARS-CoV-2 [5,6]. The causative virus of COVID-19 is SARS-CoV-2, which is the seventh member of this coronavirus family coronaviridae and the Nidovirales order [7]. Before COVID-19, only MERS-CoV and SARS-CoV coronaviruses had led to pandemics [6]. The literature reveals that around 30–80% of viral colds are due to rhinovirus and more than 15% of those are due to HCoV-229E, HCoV-NL63, HCoV-OC43, and HCoV-HKU1 [8].

Recently, the World Health Organization (WHO) declared COVID-19 a pandemic, which started in the city of Wuhan in China in December 2019 and later spread to more than 230 countries throughout the world. As of 26 September 2022, the number of positive confirmed cases of COVID-19 is 620,253,838, and it has resulted in 6,540,339 lives lost as of 22 October 2022 [8]. Human coronavirus infection is one of the top 10 fatal diseases caused in humans, such that SARS-CoV infection has a 10% mortality rate and MERS-CoV has a 36% rate of mortality [9,10,11]. The major challenge we faced during the pandemic was the development of immunity against the SARS-CoV-2 virus [11,12]. The major reason for this was the frequent mutations occurring in the genome of the coronavirus and the development of new strains with new antigenic proteins. Three months after the first case of COVID-19, on 11 March 2020, the WHO declared this to be a pandemic. The diagnosis of COVID-19 or its causative viral agent is performed through background observation of patients, clinical signs of the disease, and some laboratory-developed the technologies such as diagnostic tests including the detection and quantification of virus particles through nucleic-acid-based methodologies, antigen-antibody detections, computerized tomography (CT) scanning, enzyme-linked immunoassay (ELISA)-based methods, and blood cultures. Very recently, advanced chip-based sensors were developed for the detection of SARS-CoV-2 for long-term management. It is crucial to monitor the virus for a longer time in humans with and without symptoms. This prevents the spread of the virus [13]. The clinical signs and symptoms also vary depending on the strain and sub-strain of SARS-CoV-2 [1]. The symptoms of SARS-CoV-2 infection vary from person to person as it affects different people in different ways. SARS-CoV-2-infected patients show a very wide range of symptoms varying from very mild to severe depending on the viral load, the type of variant that caused the disease, and the immune system of the infected person. The symptoms also change over time in an infected person. There are three clusters of symptoms that have been identified. The first cluster of symptoms is associated with the respiratory system, including cough, breath shortening, and fever. The second cluster of symptoms is linked to the musculoskeletal system, including muscle pain, joint pain, headache, tiredness, and fatigue. The third cluster of symptoms is associated with the digestive system, including gastrointestinal discomfort, diarrhea, abdominal pain, vomiting, etc. [12]. In some severe pathogenic conditions, it infects the lungs and directly damages them. Lung damage is one of the very specific characteristics of HCoV infections that results in dyspepsia [14,15,16]. Due to the high variability in the signs and symptoms, the detection, identification, and characterization of coronavirus play a very important role in the treatment and management of COVID-19.

Various research groups and pharmaceutical companies are working to develop an effective and efficient anti-coronavirus vaccine and drug molecules. The vaccine development strategies include conventional vaccines, subunit vaccines, nucleic-acid-based vaccines, nanotechnology-based medicine, and nanoparticle-based vaccine delivery systems [17,18]. SARS-CoV-2 is very contagious, and its transmission is asymptomatic. Hence, vaccines are one of the best tools to protect human beings and also control coronavirus infection. Around the world, researchers and medical professionals are always trying to understand the precise viral composition, methods of infection, pathogenicity, preventative measures, immunopathogenic processes, and the most efficient therapeutic approaches [19].

## 2. Pathophysiology of COVID-19

Coronaviruses have a long 29–32 kb polymeric +ve-sense single-stranded RNA enclosed in the center of the viral particle and are surrounded by a lipid membrane with a lattice composed of 4 types of repeated structural proteins (known as coat proteins or capsid proteins), and these proteins are “spike (S) protein, nucleocapsid (N) protein, envelope (E) protein, and membrane (M) protein” (Figure 1) [9,10].

The primary target of coronavirus in the human body is the lungs. The SARS-CoV-2 virus has haphazardly arranged spike protein(s) on its capsid, which binds with the specific receptor present on lung cells called angiotensin-converting enzyme-2 (ACE2). ACE2 plays a critical role in fibrosis. The S protein of both SARS-CoV and SARS-CoV-2 binds with the human ACE-2 receptor, whereas, MERS-CoV binds with the dipeptidyl peptidase-4 (DPP-IV) receptor, mainly expressed on kidney cells [20,21]. Once viral particles enter the lung cells, they then replicate very rapidly, resulting in damage to the lung alveoli. The binding affinity of the S protein of SARS-CoV-2 with the human ACE-2 receptor is more than ten-fold higher than that of the SARS-CoV virus [22], which justifies its high human-to-human transmissibility. As soon as the epithelium of the lung is damaged by the coronavirus, proinflammatory cytokines are released as an immunological response that leads to acute respiratory distress syndrome and multi-organ damage [23,24]. There are many laboratory diagnostic matrices available that explain the immunological responses such as lymphopenia (occurring in more than 80% of cases), reductions in circulatory platelets number and serum albumin levels, and the rise in some metabolic enzymes such as “aminotransferases, lactic dehydrogenase, creatine kinase, and C-reactive protein levels” [24,25,26,27]. Regarding radiographic features, consolidations, smooth or irregular interlobular septa, and surrounding pericardial enlargement are the next most common COVID-19 indications, occurring in 68.5% of cases, followed by pulmonary lesions with bilateral ground-glass opacity [24,26]. Furthermore, 13% of patients have acute myocardial damage, and 32.8% of patients have acute respiratory distress syndrome [27].

Like SARS-CoV, SARS-CoV-2 binds to the ACE-2 receptor with the receptor-binding domain (RBD) of the spike (S) protein [28,29]. Afterward, the cell membrane of the host fuses with the host-origin lipid envelopes of the SARS-CoV virus, and viral entry into the host cell takes place. Transmembrane protease, serine 2 (TMPRSS2) along with furin initiate the activation of the spike protein, and this activation process is one of the very important steps in the viral infection, pathophysiology, and its transmission within the host body [30]. Hence, the human as a host and the cellular affinity depend on the availability of ACE-2 receptors, TMPRSS2, and furin on human cells and the protein sequences in these protein receptors (Figure 1). Many research articles explain that the number of ACE-2 receptors rises very significantly in inactive or passive smokers when compared with healthy people, and, hence, smokers are very prone to contracting SARS-CoV-2 infection, pathogenicity, and progression of the disease [31,32,33,34].

## 3. Therapeutic Strategies

Every class of people and all age groups of persons are susceptible to infection with coronavirus disease. A fresh host receives the SARS-CoV-2 virus from both symptomatic and asymptomatic patients in the form of droplets [35]. However, epidemiological analysis reveals that the SARS-CoV-2 infection rate is much less in young-age-group children, and mortality due to the SARS-CoV-2 disease is very high in the case of old-age-group populations [36,37]. After SARS-CoV-2 infection, the monocytes and macrophages in our circulation accumulate at the site of infection, and both T lymphocytes and B lymphocytes initiate strong immune responses to clean up the viruses from the host body. In a healthy human body, these immunological responses by various immune system cells work as a defense system, whereas in an immunocompromised person or a patient with immunomodulatory ailments, a strong cytokine storm/flux takes place, which results in multi-organ failure, damages many internal systemic organs, and can also be lethal to the host [38].

To date, we do not have any approved treatment strategies for COVID-19, even though we are seeing a periodical surge in positive detected cases of COVID-19 patients. Hence, the global scientific community has been working very hard for the development of a very effective and affordable vaccine as well as treatment. Drug repurposing brings a great benefit to drug delivery for new diseases or pandemic conditions [39]. At the same time, the repurposing of many existing drugs developed for various diseases showed considerable therapeutic potential against COVID-19. Herein, we discuss the various types of therapies that have been developed for COVID-19 treatment [1].

### 3.1. Antiviral Drug Molecules

To date, the most accepted and adopted treatment strategy is the use of antiviral drug molecules and interferons. Some of the antiviral drug molecules that have been recommended for the management of COVID-19 are the viral protease enzyme inhibitor lopinavir, ritonavir, and the nucleoside analog ribavirin. These molecules are used alone or as a combination of protease inhibitors and nucleoside analogs. Another antiviral drug molecule is remdesivir, which was initially developed against the Ebola virus when the Ebola pandemic occurred in African countries, and chloroquine and hydroxychloroquine, which have been developed and used as antimalarial drugs molecules and are used to treat coronavirus disease. Remdesivir is chemically a nucleoside analog and works as a prodrug molecule. Its mechanism of action affects the working of the viral RNA polymerase enzyme, and experimental data show its suppression of SARS-CoV-2 viral infection in Vero cell lines [40,41]. The use of remdesivir in clinically ill patients has shown mixed results [41,42]. Antimalarial drugs chloroquine and hydroxychloroquine have also shown their effectiveness against SARS-CoV-2 in in vitro experiments (Figure 2). However, in a randomized clinical trial study in China, hydroxychloroquine did not show any significant improvement in the patients who were already on the conventional treatment. Rather, such patients developed an additional serious defect [43].

### 3.2. The S Protein and ACE2 Interaction Inhibitors

The spike (S) protein of SARS-CoV-2 is the protein that recognizes and binds with the host cell receptor ACE2. Any drug molecules such as Arbidol, proteins, or any other category of substances that inhibit or slow down the binding of the S protein with the ACE2 receptor may prevent the entry of the virus into a host cell and ultimately the spread or transmission of viral particles from host to host (Figure 2). As per the current research findings, the coronavirus fusion inhibitor peptide EK-1 significantly reduces the receptor-mediated entry of viral particles into the host cell and also stops the formation of six-helix bundles that form due to the interaction of specific heptad repeated sequence-1 in the S2 components of the S protein of the virus with the host cell [44].

### 3.3. Neutralizing Antibodies

Since monoclonal antibodies, unlike vaccinations, offer instant immunity, giving out pure monoclonal antibodies with neutralizing capacity may be another way to treat SARS-CoV-2. Currently, the discovery of efficient neutralizing antibodies predominantly focuses on the S protein bound on SARS-CoV-2. SARS-CoV-2 can cross-react with two strong camelid single-domain antibodies against SARS-CoV and MERS-CoV, breaking the ligand-binding surface [45]. The potential for the 47D11 hu-mAb to attach to the S protein RBD to counteract SARS-CoV-2 infection was recently established [46]. The SARS-CoV monoclonal antibody, also known as the S309 antibody, effectively inactivates SARS-CoV-2 by interacting with the S protein [47]. As a result, using different monoclonal antibody combinations that can concurrently engage the specified non-RBD and RBD can be a great option for the safe and effective treatment and prevention of COVID-19.

### 3.4. Immunotherapy

A tight connection exists between ARDS and the excessive cytokine serum levels (cytokine storm) that cause considerable organ damage in COVID-19 patients who are critically unwell. To stop the spread of COVID-19, cytokine storm treatment and prevention may be a useful approach. Clinical research has demonstrated that a rise in IL-6 levels is the primary driver of inflammation [37]. Through its interaction with gp130, the complex formed when IL-6 binds to the soluble IL-6 receptor (sIL-6R) or membrane IL-6 receptor (mIL6R) triggers the inflammatory response. Tocilizumab (a monoclonal antibody against IL-6) preferentially acts on sIL-6R and mIL-6R to prevent the signal transduction that initiates inflammatory responses [48]. A recent study found that hydroxychloroquine and chloroquine can inhibit the production of proinflammatory cytokines, including IL-6, which are responsible for the creation of cytokine storms [49]. However, cost and safety concerns may prevent the use of tocilizumab in COVID-19 treatment. According to a recently released study, sarilumab, another IL-6 receptor antagonist, may help extremely ill COVID-19 patients recover quickly by reducing inflammatory processes [50].

### 3.5. Convalescent Plasma Therapy

Convalescent plasma (CP) therapy is another efficient treatment; even so, to ensure high neutralizing antibody titers, CP must be administered at least two weeks after recovery [51]. A recent study found that serum from several different patients could neutralize SARS-CoV-2 that was obtained from patients with severe respiratory diseases [52]. In a different trial, it was anticipated that five COVID-19 patients who were given CP including neutralizing antibodies would show an improvement in their clinical condition [53]. Numerous clinical trials continue to explore CP as a COVID-19 treatment option at present.

### 3.6. Preventive Vaccination Strategies

A global multifaceted strategic approach to vaccine development is being attempted to create efficient SARS-CoV-2 vaccines. There is a chance for quick vaccine development because the genomic and structural information on SARS-CoV-2 has become available much faster than that on other HCoVs [28,45,54,55]. Additionally, information gathered from research on SARS-CoV and MERS-CoV vaccine development is useful for creating a vaccine candidate for SARS-CoV-2 [56,57].

### 3.7. Inactivated or Live-Attenuated Vaccines

Vaccines that are denatured or live-attenuated offer benefits including activating pattern recognition receptors and having high immunogenicity (Figure 3). The viruses can spread and are living, yet they are not harmful. Long-term surveillance is necessary to evaluate the safety of the vaccine, however, due to the possibility of live viruses. The first clinical trials of many inactivated viral vaccines against SARS-CoV-2 have just started at Sinovac Biotech in Beijing, China. More recently, synthetic genomics was used to create recombinant SARS-CoV-2 from viral DNA fragments [58,59]. These results open the door to the development of live-attenuated vaccines against SARS-CoV-2. In addition, Codagenix, Farmingdale, NY, USA is investigating potential SARS-CoV-2 vaccine candidates employing a “codon-optimized off” virus attenuation method [60].

### 3.8. Recombinant Vaccines

Through genetic engineering, recombinant vaccines enable live viruses to retain some extra genes derived from pathogens, translating the target protein and inducing the correct immune response (Figure 3) [61]. Recombinant vaccines have the advantages of a sufficient expression of the target protein, long-term stability, and the activation of potent immune responses [62]. Based on studies that have demonstrated that they can trigger extremely potent immune responses to foreign antigens, vaccines based on vaccinia viral vectors are currently being examined for use in numerous clinical trials [63,64]. The availability of a large-scale manufacturing technique, such as in the example of when Bavarian Nordic A/S created and gave significant quantities of its own smallpox vaccine IMVAMUNE^®^ to the US government, could be another benefit of a vaccinia virus-vector-based vaccine. We can use a variety of adenovirus (Ad) vectors to our advantage due to their broad-spectrum viral affinity and infectivity in dividing and non-dividing cells. Human Ad serotype 5 (Ad5), which may be easily produced at high titers among the human Ad sera found so far, is the most extensively researched gene transfer agent [64,65]. The drawback of Ad vectors is that many individuals who have already been exposed to the Ad serotype have developed pre-existing immunity to it. The single-stranded DNA virus known as adeno-associated virus (AAV) is non-pathogenic, immunogenic, and enclosed within a carrier. AAV has both the benefits and characteristics of Ad vectors. Due to their poor titer production efficiency, AAV vectors require a very effective large-scale production technology, such as the baculovirus system [66,67]. Ads’ traits and benefits are present in AAV as well. In a phase I/II COV001 experiment, ChAdOx1 nCoV-19 (AZD1222), an Ad-based recombinant vaccine created at the University of Oxford in Oxford, UK, was discovered to be resistant, and a potent immune response to SARS-CoV-2 was produced in all participants [68]. Nearly all the patients who received AZD1222 displayed a four-fold increase in the antibody’s ability to neutralize the SARS-CoV-2 S protein [12]. Additionally, AZD1222 usage has not been associated with any negative side effects of note [68].

## 4. Application of Nanotechnology in COVID-19 Therapeutics

As we already know, there are no therapeutic drugs available for the management of COVID-19 that are approved by the FDA. Hence, a rapid and point-of-care diagnosis of coronavirus is very significant and helpful in the timely management of coronavirus infection. In the diagnosis of coronavirus disease, nanotechnology-based methodologies play an important role so that the spreading of coronavirus can be stopped. Researchers working in the biomedical field have been investigating the relationships between high infection rates as well as the capacity of distinct nanostructured materials and viral vectors to deliver genes. In COVID-19 management, the nanotechnology-based approach has a wide range of uses and can act at various phases of the disease. It has the potential to block virus–cell contact, lipid bilayer blending, the endocytosis of cells, transcription, translation, and viral reproducibility in addition to stimulating subcellular processes that inflict irreparable harm on viruses. To create delivery methods that may be applied in several disciplines, nanotechnology scientists have investigated the biological mechanisms of gene carriers [69,70]. Since viral particles and nanostructures operate in a similar dimension, developing vaccines and performing immune engineering relies heavily on this strategy (Figure 4). Nanopharmaceuticals may be the ideal substitute for cutting-edge vaccine development technologies since nanoparticles (NPs) are tools that may mimic the functional and structural characteristics of viral infectious particles [71,72,73]. Nanosensors and medication delivery for various diseases have already demonstrated the value of nanotechnology and nanoparticles. Additionally, nanotechnology can give a more comprehensive perspective of new vaccine design approaches. For example, a unique nanoparticle-based vaccine metastatic platform, helpful nanomedicines for treating SARS-CoV-2 infections, and a nano-based formulation for SARS-CoV-2 therapies are all being produced. To quickly identify and create suitable nanovaccines and therapeutic options, including novel nano-based technologies, scientists have been working hard up to this point. We will talk about COVID-19 management alternatives in nanomedicine in this review. To do this, we will first look at the pathology of COVID-19 to provide the groundwork for uncovering any openings or loopholes in the pathology of this virus where the application of nanotechnology can be useful. In the current tragic condition wherein coronaviruses and their sub-strains are major challenges to researchers and scientists, especially in vaccine development, nanotechnology, nanopharmaceuticals, and nanomedicine development provide good, effective, and comparable techniques [74,75].

Since the emergence of the COVID-19 outbreak, nanoparticles of selenium (Se) have gained attention due to their capability to manage COVID-19 infection. Nano-Se has excellent antiviral efficiency with the potential to neutralize and eradicate SARS-CoV-2 variants. However, further research needs to be investigated [68].

### 4.1. Theragnostic Nanoparticles

At present, nanotechnology plays a major role in the development of groundbreaking products in short periods of time with great sensitivity and selectivity to diagnose and treat various diseases including COVID-19. The advantages associated with nanoparticles that make them suitable and make it possible to use nanoparticle-based approaches and overcome various challenges faced during the use of conventional medicines are the nanoscale size, their very low levels of toxicity, their electrical charges, their chemo plasticity behaviors, etc. The binding of corona viral particles with host cells, their entry, and various steps of the replication process can be targeted using nanoparticle-based approaches. The RBD of the spike (S) protein and the complete S protein itself is the main component that binds with the host cell receptor and allows the coronavirus to enter the host cell. Many studies have been published wherein the binding of the S protein with the ACE2 receptor of the host cell has been blocked by a nanoparticle-based approach, and, finally, the virus entry was also stopped. After the entry of nanoparticles-mediated approaches in the management and treatment of common viral diseases, many virus infections and their pathophysiologies have been stopped by using nanoparticles and nanomedicines such as influenza virus-A and influenza virus-B [76], Ebola virus [77], human immune deficiency virus (HIV)-1 and -2 [78,79], Herpes simplex virus (HSV)-1 and -2 [80,81], hepatitis virus (B and C) [82,83], and human norovirus (HuNoV) [84]. Dexamethasone in the form of nanoparticles has been used in the treatment of COVID-19. The mechanism of action of this nanoparticle-based approach is in their anti-edema and anti-fibrotic actions. Nanoparticles have also been used in the targeted delivery of dexamethasone nanomedicines [85].

### 4.2. Intranasal Delivery Therapy

We know that the route of entry of SARS-CoV-2 into the human body is through the mucosal membrane, and due to this, many scientific groups and pharmaceutical industries have worked on the development and improvement of the targeted delivery of nanoparticles and nanomedicines into the mucosal membrane of the nasal chamber, which makes this one of the safest and effective approaches for drug delivery against any viral disease [86]. The most crucial method for addressing these pathogenic disorders is mucous therapy since SARS-CoV-2 first infects the mucosal membrane of the cornea or sinus passages. Mucosal delivery of nanoparticles is not only a safe and effective approach but also a non-invasive approach for drug delivery [87]. When determining the best delivery strategy to the nasal cavity, it is crucial to consider a nanoparticle’s characteristics, including its surface charge, size, structure, and shape, since they are essential in order to achieve a successful and secure treatment approach [88]. Small laboratory experimental animals have been used in research to investigate the mechanism used to deliver nanoparticles and nanomedicines to the lungs through the nasal passage. However, conclusions from such experiments on rodents cannot be simply extrapolated to humans. Three different forms of nanoparticles (organic, inorganic, and virus-like nanoparticles) have so far been developed with successful therapeutic delivery capabilities; they can also be given intranasally.

### 4.3. Organic Nanoparticle-Based Treatment Strategy

The important properties that make organic nanoparticles suitable for vaccination against SARS-CoV-2 are their site-specific drug targeting capabilities, controlled drug delivery, renewability, biocompatibility, and non-toxicity. Extra-cellular vesicles (or exosomes), polymeric and lipid-based NPs, dendrimers, liposomes, and nanomicelles are some of the most popular organic NPs [89,90]. The human cell membrane is composed of phospholipids, and hence lipid nanoparticles are highly biocompatible with the plasma membrane and human cells. These nanoparticles are selected in the field of biomedicine due to their biocompatibility only. There are various types of lipid nanoparticles available, but among all these nanoparticles, liposomes are the most preferred nanoparticle for intranasal uses. Liposomes are essentially a subtype of nanoparticles comprising a hydrophobic tail and a hydrophilic head constituting a phospholipid membrane [91]. The merits and demerits of the use of liposomes and other nanoparticles have been summarized in Table 1 and Figure 5. Silica nanoparticles have been used for the transportation of an antiviral chemical component, ML336, which works against the “Venezuelan equine encephalitis virus”, into VEEV-infected experimental mice. ML336 alone, without being packed into a silica nanoparticle, is not very stable and is strongly hydrophilic against this virus. The packing of ML-336 into silica nanoparticles improves the cycle time, viral titer, and bioavailability [92]. Occasionally, some nucleic-acid-derived drug molecules, such as siRNA, which is very unstable in the circulatory system, are delivered to specific target organs. This nanoparticle packaging increases bioavailability, increases stability, and reduces any kind of damage to blood circulation [93]. Another option for targeted drug delivery is the use of polymeric nanoparticles because, according to specific applications, their characteristics and roles can be manipulated [94]. Nanoparticles composed of chitosan when conjugated with therapeutic molecules increase the mucosal layer permeation of the drug molecule and also the persistency of the nanoparticle in the mucosa [95].

By removing the possibility of toxicities, antibody–drug complexes containing auristatin are utilized to treat leukemia in a reasonably safe way. However, this approach has the serious drawback of having too little drug payload. To overcome this restriction, researchers used the polymeric nanoparticles approach to create monomethyl–auristatin E nanostructure conjugates, which have great safety and enable the availability of a large amount of auristatin payload. Additionally, when the Aurora B kinase inhibitor AZD2811 was enclosed in Accurin polymeric nanoparticles, which have been demonstrated to have substantial side effects in phase two clinical trials, the toxicity was found to be significantly reduced, and the effectiveness was shown to be increased [96].

Dendrimer nanoparticles actively interact with viruses. The system that emerges enhances the antiviral activity and effectively shields the body against infection. Additionally, successful reports of dendrimer nanoparticles being employed to treat viral infectious diseases including HIV and the flu virus have been documented. The benefits and drawbacks of using polymeric nanoparticles and dendrimer nanoparticles are outlined below [97,98,99].

### 4.4. Inorganic Nanoparticle-Based Treatment Strategy

The characteristic features of inorganic nanoparticles which make them suitable for vaccine development for COVID-19 are luminosity; variable shapes, sizes, and compositions; a high surface-to-volume ratio; and the capacity to reveal numerous engagement sites across a surface. The easy induction of an immune response mediated via antigen-presenting cells using gold nanoparticles makes them a desirable candidate for use in the creation of vaccines. The ability to be easily modified for distribution through the nasal cavity is a benefit of using gold nanoparticles [100]. By spreading to the lymph nodes, they also have the benefit of triggering the immunological response linked to CD8+ (cytotoxic) T cells [100].

### 4.5. Virus-like Nanoparticle-Based Treatment Strategy

Virus-like particles (VLPs) are one class of subunit vaccines that are composed of non-infectious components of viral particles along with adjuvants. VLPs can behave like a non-infectious virus, but they lack the genetic material of the virus and hence do not cause any disease. Sometimes the structural proteins from different viruses are combined to make recombinant VLPs (Table 1). Higher immunogenicity can be achieved by using virus-like nanoparticles to produce a potential immunogenic epitope. Additionally, because virus-particle-like nanoparticles are tiny, they can act as adjuvants, and altering adjuvants can trigger an immune response that is much more powerful than a virus [101,102,103]. It has been found that virus-particle-like nanoparticles work as a vaccine by producing a very large number of T cells and antibodies that can trigger various sorts of immunological reactions in order to increase immunity and prevent additional infection as a result of the intranasal delivery of virus-particle-like nanoparticles using the influenza virus [103]. Earlier research has suggested that cell membrane nanovesicles and exosomes can bind and destroy bacterial toxins [104,105,106]. Additionally, the biomimetic biosynthesis method, which involves the synthesis and display of proteins on the cell surface, has recently been used to create cell plasma membrane nanovesicles containing proteins with the same structure and activity as native cells. Plasma-membrane nano-vesicles are nano decoys that can compete with host cells for virus and cytokine adsorption because they are designed to exhibit elevated levels of ACE2 and an abundance of cytokine receptors. According to research, a nano decoy effectively binds and neutralizes inflammatory cytokines including IL-6 and GM-CSF and greatly reduces the multiplication and infestation of coronavirus [107,108]. Accordingly, an effective alternative to SARS-CoV-2 and cytokine storms is a therapeutic technique utilizing biological membrane nanovesicles. Exosomes are developing nanomaterials in current cell regeneration, therapeutic, and diagnosis studies. They are tiny nanovesicles with a size of 30 nm to 150 nm that are released in all forms of cell-to-cell interactions [108]. Exosomes carrying the S protein of SARS-CoV have been shown to enhance the development of neutralizing antibodies by first stimulating only with the S protein vaccine and afterward boosting with an adenovirus vector vaccination [109]. Exosome-based treatment for SARS-CoV-2, hence, has the potential to be sufficiently implemented.

### 4.6. Pulmonary Delivery Using NP Inhalation Aerosols

If the advantage of drug delivery through the nasal cavity is to act on the mucous membrane area where the infection occurs, then the lungs are an important organ for drug delivery because they are another target for treating SARS-CoV-2 infection, which infects primarily through the respiratory tract (the upper airways and lungs) [110]. Therefore, the use of inhaled aerosols is suggested as an effective non-invasive mode of administration. Additionally, the delivery of inhalable nanoparticles into the lungs overcomes disadvantages, such as side effects caused by high drug concentrations in the serum with conventional oral or intravenous drug administration methods. Various nanotechnologies have been applied to develop nanoparticles that can function as lung inhalation aerosols. These respirable nanoparticles can be encapsulated by microparticles manufactured down to five microns to fit the aerodynamic size range or agglomerate into an aerodynamic size range. Most nanoparticles are delivered directly into the lungs either via spraying colloidal dispersions or dry powder inhalers and pressurized metered-dose inhalers in solid forms [111].

Until the present day, lipid nanoparticles are one of the most studied and commonly used nanoparticles for the safe and efficient delivery of medicines in the mucosal system or pulmonary system [1]. One of the unique features these nanoparticles have is that these nanoparticles are produced from commonly available substances in lung-like surfactants, and this makes these nanoparticles the prime candidate for drug delivery vehicles for the lung or pulmonary system [112]. Liposomes are mainly composed of lipids, and commonly used nebulizers are used to deliver these liposomes into the respiratory system and mucosal system of the body [113]. There are some disadvantages associated with the nebulized-mediated delivery of liposomes, such as in the stability of the drug molecules and leakages in the nebulizer. To overcome these disadvantages, many attempts have been undertaken, and one of the successful attempts was to develop a dry powder form of the liposome–drug formulation [114,115,116]. One of the advantages associated with cationic liposomes is their auto assembly with their negatively charged nucleic acids, and, hence, these liposomes are commonly used in gene delivery systems for the respiratory system and more specifically for the lungs [117]. Cationic liposomes have also been studied and used for the delivery of large high-molecular-weight peptides [118]. Polymeric nanoparticles also share advantages such as an increment in drug stability and a slow and steady release of drugs into the lungs and mucosal system. In clinical applications, cationic lipid nanoparticles have more advantages over polymeric nanoparticles, but the application of cationic polymeric nanoparticles in drug delivery to the lungs cannot be discarded [119,120,121].

## 5. Nanotechnology-Based SARS-CoV-2 Vaccine Development

Besides traditional vaccine modalities (i.e., inactivated vaccines, live attenuated vaccines, and recombinant protein vaccines) and DNA- and vector-based vaccines, nanoparticle vaccines offer a unique opportunity to advance vaccine science and provide tractable solutions to the current pandemic and beyond [122,123]. Nanoparticles are adjustable, nanosized granular entities that resemble the architectural characteristics of naturally occurring pathogens. Since they provide mechanisms to trigger potent nAb reactions or wider antibody-based immunization, which may target a variety and development of viral pathogens, they are very attractive technologies for the production of next-generation vaccines. There are currently 60 more possibilities in various phases of preclinical studies, with at least 26 nanotechnology vaccination alternatives having moved into clinical trials with humans (Table 2).

Such vaccines are available in a range of formats, including liposomes, protein nanoparticles, micelles, and nanoparticles that resemble viruses. Nanomaterial-based vaccines can indeed be divided into two groups depending on the protein packing techniques: [1] those that encapsulate vaccine proteins or nucleic carriers within the cores, and those that display vaccine antigens on their surfaces. Nanoparticles showcasing vaccine allergens can engage antigen-presenting cells (APCs) and/or effectively promote B cell binding site (BCR) cross-linking, leading to powerful immunogenicity.

There are many nanotechnology-based COVID-19 vaccines that have been developed, and these vaccines can be categorized into three main classes based on their functional components: (a) virus particles, (b) viral protein-based, and (c) nucleic-acid-based [124,125,126]. Nanoparticles have manipulated and controlled properties and are nanoscale-sized particulate entities that resemble the structural characteristics of natural viruses. Since they provide mechanisms to trigger potent nAb responses or broader antibody-based immunity that may better account for the variety and evolution of viral infections, they are very attractive platforms for the development of next-generation vaccines. Biocompatible polymers are typically used to create nanocarriers because they have the best properties for blood–brain barrier administration, including safety, high stability, the capacity to load a variety of therapeutic chemicals, control over release kinetics, and ease of chemical modification. Targeted drug delivery has benefited greatly from the development of nanocarriers made of nanoparticles, micelles, dendrimers, polymeric nanoparticles, and polymeric or lipid-based carriers such as liposomes. An effective adaptive immune response that produces immunological memory is the aim of a reasonable nanoparticle-based vaccine design. This denotes a particularly long-lived monitoring system of specialized memory B lymphocytes that can accumulate a significant number of plasma blasts through large clonal expansion before differentiating into plasma cells capable of generating antibodies in response to pathogen exposure. These antibodies should prevent the virus from interacting with the host cell (neutralization), which will eventually result in virus eradication. In addition to the production of neutralizing antibodies, the removal of virus-infected host cells depends on the activation of cytotoxic CD8+T cells (CD, cluster of differentiation). The most popular delivery method for COVID-19 NVs is intramuscular (IM) administration since it both results in minimal local reactogenicity and provides optimal bioavailability, which increases the immunogenicity caused due to the nanoparticle-based vaccine’s overall immunogenicity. Nanoparticle-based vaccines enter the lymphatic channels upon IM injection and are then transported to the lymph nodes. Muscle cells of the host body and some important antigen-presenting cells take up the nano vaccines. Irrespective of the nanoparticles used, the aim of these nanoparticle-based vaccines is the delivery of these antigenic components enclosed in nanoparticles to the host cells, their processing and presentation of processed antigenic components on the cell surface with the help of a major histocompatibility complex, and, lastly, the recognition of this MHC with the T cell receptors and activation of the B cells for the production of antibodies.

### 5.1. Subunit Vaccines

Candidates for subunit vaccines must significantly improve immunogenicity by inducing an immune system response when taken together with the adjuvant components employing structural elements of coronavirus. Hence, it is of utmost importance to create a vaccine that selectively targets the SARS-CoV-2 S polypeptide component. This is due to the S protein’s combination of membrane fusion and receptor-binding domains [108]. Antibodies that are activated by vaccines developed by using the S protein block viral attachment and entry by preventing virus binding and, later, lipid membrane conjugation [127]. The spike (S) protein of coronavirus, which is the target binding site for ACE2, is an important tool and candidate molecule for both therapeutic and prophylactic vaccine development [128] Along with this, nanoparticles equivalent to immunogenic viruses have been produced with the help of recombinant nanoparticle-based vaccine development (Novavax^®^ proprietary, developed by Novavax, Gaithersburg, Maryland, United States) and the spike (S) protein of SARS-CoV-2 [129]. A novel SARS-CoV-2 protein subunit vaccine has been developed by the University of Queensland, Australia, employing a “molecular clamp” technique, which prevents viral spike proteins from adhering in the first step [130]. Alternatively, there is a lot of work being undertaken to create protein subunit vaccines employing nanotechnology-like proteins and viral-like particles. It has been discovered that receptor-binding domains in SARS-CoV-2 have a stronger binding capacity for ACE2 than receptor-binding domains in SARS-CoV [131]. Hence, a receptor-binding-domain-based SARS-CoV vaccine can assist in the prevention of SARS-CoV-2 binding and is essential for the development of SARS-CoV-2 vaccines. Additionally, a large number of research institutions and international pharmaceutical corporations are actively working on developing RBD-based vaccines, which are efficient in both prophylactic and therapeutic techniques. Another benefit of vaccinations based on RBDs is that they lessen host immunity augmentation [127].

### 5.2. Nucleic Acid Vaccines

As soon as SARS-CoV-2 enters the host cell through the binding of ACE2, the viral protein is expressed inside the host cell and induces cell-mediated immunity. This is the basic principle for the development of all nucleic-acid-based vaccines, and as nucleic acids are the least immunogenic, immunization of a host with a nucleic-acid-based vaccine, hence, does not directly cause any adverse reactions. Nucleic-acid-based vaccines are among the important and effective immunization methods whereby we use a laboratory-produced nucleic-acid-based vaccine to activate the immune system in the same way as it occurs in the case of live attenuated vaccines. The possible benefits of mRNA vaccines are due to their enhanced immunostimulatory qualities, which imitate the pathogenic mechanism. A single vaccine has several mRNAs combined to enhance the effect [131]. One of the very well-developed and used mRNA-based SARS-CoV-2 vaccines is developed by Moderna, Cambridge, MA, USA, named mRNA-1273. This type of vaccine is composed of artificially produced mRNA, which expresses the spike (S) protein inside the host cells and activates host immune responses [132]. After vaccination via muscular injection of the vaccine, the SARS-CoV-2 S protein triggers a particular antiviral immune response. In contrast to traditional vaccines consisting of smaller components of living or attenuated viruses, nucleic acid vaccines do not need viruses to be created. As a result, because safety is assured, only the completion of the mRNA-1273 phase I study is necessary to enable a speedy advancement in the ongoing investigation of effectiveness [132]. A lipid nanoparticle platform served as the foundation for the construction of mRNA-1273, but novel nanoparticles are now being used to transport oligonucleotide vaccines more effectively. Lipid nanoparticles, dendrimer nanoparticles, and polymeric nanoparticles are used for their efficient distribution and excellent persistence in the case of mRNA-based immunizations. A codon-optimized mRNA vaccine encoding the SARS-CoV-2 receptor-binding domain [133], BNT162b1, is being developed by Pfizer, New York, NY, USA. There are many other mRNA-based COVID-19 vaccines that are enclosed in lipid nanoparticles that have been developed and are listed in Table 3.

To improve immunogenicity, these vaccines employ an RBD antigen that has the trimerization domain of T4 fibrin added. As soon as the initial gene sequence was made public [134], the Coalition for Epidemic Preparedness Innovation started working on vaccinations in collaboration with a group researching vaccines on a cutting-edge platform. Because of this, the mRNA-based SARS-CoV-2 candidate advanced to the level of human clinical trials [17]. In addition, among the nucleic acid vaccines is INO-4800, a potential DNA vaccine created by Inovio Pharmaceuticals, Inc., in Plymouth Meeting, PA, USA. INO-4800, a nucleic acid vaccine, functions similarly to RNA vaccines in that it can elicit an immune system response by being transcribed into proteins in human cells. In terms of production costs and purifying techniques, nucleic acid vaccines have a lot of advantages over traditional vaccines. The generation of abnormal proteins, which can happen with transgenic vaccines, is additionally avoided with the nucleic-acid-only structure [135]. Nevertheless, the quantity of plasmid introduced into the organism, as well as the ideal injection time and channel, has a significant impact on the effectiveness of nucleic acid vaccines. To enhance the effectiveness and durability of the administration of nucleic-acid-based vaccines, NPs such as cationic liposomes, dendrimer nanoparticles, or polymeric nanoparticles have been used [133].

### 5.3. Nanoparticle-Based Vaccines

MERS-CoV, as opposed to SARS-CoV, has been used numerous times to incorporate nanotechnology-based approaches into drug discovery and development or vaccine development. Significantly, it has recently been demonstrated that virus-like particles are appropriate for producing vaccines or therapies for MERS-CoV-infected characteristics. Compared with other tiny vaccines, nano-sized virus-like particles have the benefit of being more effectively administered through the lymphatic and microvascular system [136]. Additionally, they have the benefit of lessening the body’s chronic inflammation and, like pathogens, have an easy time entering cells. Additionally, the introduction of numerous antigens enhances the efficiency of the antigen-presenting cell. The increased responsiveness and effectiveness of the vaccine as a result of the manufactured combination recognized by the T cell receptor ensures the safety of patients [136]. Nano-sized virus-like particles that reach the human host are directly involved in B signal transduction and the body’s immune stimulation [136,137]. The properties of such artificial submicron virus-like particles are fundamental to creating vaccination systems. In animal experiments, nano-sized VLPs, virus-like particles, have also been found to successfully combat pathogens by boosting the immune system response [137]. Recently, bombyx mori caterpillars were used to create the MERS-CoV S peptide. The submicron VLPs that displayed natural structural epitopes made through treatment with detergent and cellular membranes were then subjected to this [138]. A further experiment used a continuous compaction of red blood cells through a 1 m filter to create submicron VLPs that could function as lipid nanoparticles in red blood cells [139]. Synthetic S, transmembrane, and envelope proteins were used to create MERS-CoV submicron virus-like particles, which were then examined in animal models and found to demonstrate higher responsiveness [101]. Nano-sized virus-like particles have a broad range of uses, can improve vaccine safety and efficacy, and have a lot of benefits that can be put to good use for certain uses [140,141]. These results can be used to manage SARS-CoV-2 infection because they were generated for the spike (S) protein, which is typically seen in MERS-CoV and SARS-CoV.

## 6. Deactivation of Coronavirus Outside Host Cell with the Use of Nanotechnology

Coronavirus is easily attenuated by UV light, extremely alkaline environments, and acids. It is activated at temperatures between 1 and 35 °C [142]. Additionally, SARS-CoV-2 is quickly inactivated by routinely available disinfectants, and its level of stability varies significantly depending on the elements that make up the infectious particle’s surface [143]. SARS-CoV-2 is activated similarly to SARS-CoV, in aerosols and on surfaces; hence, surface treatment with nanoparticles that are effective against SARS-CoV will be sufficient for SARS-CoV-2 [35]. Nanotechnology has the potential to offer alternatives to standard disinfection processes for viruses used in public or medical settings, which typically rely on chemical, physical, and biological techniques, that are more successful. Additionally, the release rate of metal ions, which are antibacterial, can be easily controlled by utilizing NPs on the outside surfaces of things that need antibacterial action. Due to their ability to aggregate in cells, nanoparticles can overcome the drawbacks of antibacterial agents or metal ions that readily escape from cells. Currently, paintings and food trays are treated with silver, which has been utilized as an antibacterial agent since ancient times. It has already been demonstrated that silver nanoparticles (Ag nanoparticles) have antiviral properties against a variety of viruses. Ag nanoparticles [144] work as antiviral agents by dissolving and releasing Ag+ ions that are harmful to microorganisms. The function of proteins that are crucial for virus reproduction, such as enzymes requiring thiols, can be disrupted by Ag+ ions, which can interact with proteins on the surface of a virus or invade and concentrate in host cells [145,146]. Ag nanoparticles may also have an additional antiviral effect where, based on their size, they effectively obstruct virus binding to host cells through their physical interaction with the viral envelope [146]. It has been discovered that, in comparison with particles of different sizes, Ag nanoparticles with a size of around 10 nm exhibit the highest physical interaction and antiviral activity [147]. Additionally, after attaching to the virus surface, Ag nanoparticles have an antiviral impact by destroying the virus structure by releasing reactive oxygen species (ROS). Medical equipment already uses Ag nanoparticles and has them applied to it. They can be utilized to inactivate SARS-CoV-2 via the antiviral impact of Ag+ ions when added to face masks and air filters [148]. Currently, it has been observed that coating filters with Ag-nanoparticles successfully block the bacteriophage MS2 from dust.

Another metal ion that has been recently proven to have an antiviral effect against HuCoV-229E is copper, and it is hoped that it may have anti-COVID-19 effects as well [149]. The incubation of viruses on a copper-coated surface deactivates their nucleic acids and the viruses are neutralized [149]. The possible mechanism of action of copper is mediated by inactivating the physiological functions of viral proteins through the production of hydroxyl radicals, which ultimately neutralize virion particles [150]. Similar kinds of encouraging results were also reported for the inactivation of SARS-CoV-2 on a copper-coated surface [35]. Not only in action but also economically, copper is better than silver for its antiviral uses and can very easily be used to produce polymeric nanoparticles. Copper-based nanoparticles also have shown very good stability when compared with other nanoparticles, and due to all these reasons, copper- or copper-oxide-based nanoparticles are some of the best strategies for neutralizing SARS-CoV-2 outside the host body. One of the applications wherein it was found to kill the influenza virus was by using a mask with copper-oxide-based nanoparticles [151]. Viral inactivation is effectively accomplished by metal nanoparticles and graphene derivatives (GDs and nanoparticles) [152]. As part of the graphene derivatives antiviral strategy, the negative charge on the coated surface of a graphene derivative aids its adhesion to positively charged virus particles [153]. When added to antibodies against viruses via nanotechnology, graphene derivatives have a remarkable impact on rotavirus and influenza virus infections [154,155]. Additionally, recent studies have demonstrated that this characteristic of graphene derivatives can be exploited to treat, identify, and prevent SARS-CoV-2 [155]. Numerous studies have already demonstrated that iron oxide nanoparticles have antimicrobial properties. The US Food and Drug Administration (FDA) has also approved iron oxide nanoparticles for the treatment of anemia due to their remarkable biocompatibility [156]. Recent studies have demonstrated the interaction of iron oxide nanoparticles with the S protein of the SARS-CoV-2 virus as well as the potential antiviral activity of iron oxide nanoparticles [157]. The reactive-oxygen- and free-radical-producing capacity of iron oxide nanoparticles may be one of the important tools for neutralizing the SARS-CoV-2 virus outside the host body [158,159].

## 7. Conclusions

Since SARS and MERS infection rates are substantially lower than those of COVID-19, therapy and vaccine candidates for these diseases have not previously undergone thorough study and development. Concerning its clinical characteristics, COVID-19 does not appear to differ significantly from SARS. However, it has a fatality rate of 2.3%, which is significantly less than that of MERS (34.4%) and SARS (9.5%). Nanotechnology-based research and development are crucial to put an end quickly and successfully to this outbreak. The development of SARS-CoV-2 therapies and vaccines may advance because of the cumulative advances in these virus-fighting nanostructured materials. With the aid of nanomedicine, the arduous COVID-19 pandemic, which has not yet ended, is currently advancing in the direction of eradicating the virus gradually. Several businesses are currently eschewing conventional SARS-CoV-2 prevention and treatment methods in favor of employing nanotechnology to create a range of vaccines and medicines and perform clinical trials. Dexamethasones, for instance, a COVID-19 therapeutic drug that has been introduced via various nanoformulations, has significantly changed how COVID-19 is treated [85]. Additionally, the successful completion of phase three clinical trials for the Pfizer-developed liposome mRNA vaccine (BNT162b) might be regarded as a major accomplishment for nanomedicine [159]. Additionally, technology that quickly and cheaply detects SARS-CoV-2 by applying gold nanoparticles [160] and diagnostic technology that can deactivate SARS-CoV-2 in the external environment by using nanomaterials such as silver nanoparticles have been developed [144,145]. Nanoparticles with Cu or CuO [150] and GDs [152] are both contributing to the prevention and control of COVID-19. However, it is thought that the current platform needs to be updated globally for research in numerous sectors to be more effective because of the complex scenario brought on by COVID-19. Therefore, alternatives to this shift in the research and development paradigm, such as nanotechnology and nanomedicine, may be appropriate.

## Figures and Tables

**Figure 1 pharmaceutics-15-00451-f001:**
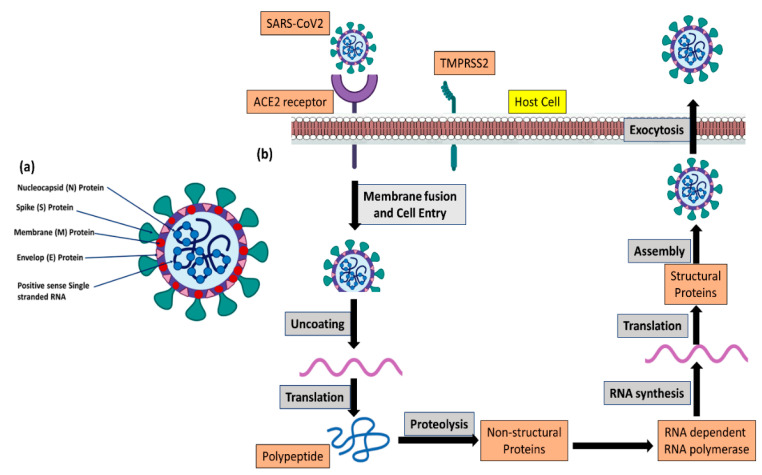
Coronavirus structure and replication steps in host cells. (**a**) Schematic diagram of SARS-CoV-2. (**b**) Schematic representation of the replication cycle of SARS-CoV-2.

**Figure 2 pharmaceutics-15-00451-f002:**
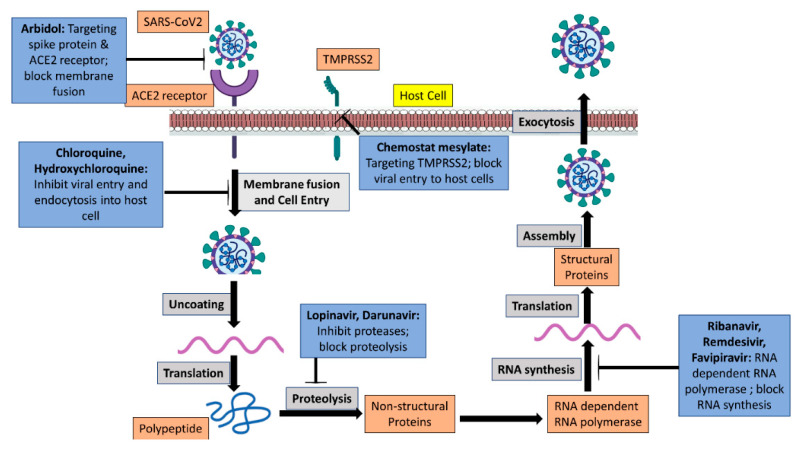
Antiviral drugs and their mode of action against SARS-CoV-2.

**Figure 3 pharmaceutics-15-00451-f003:**
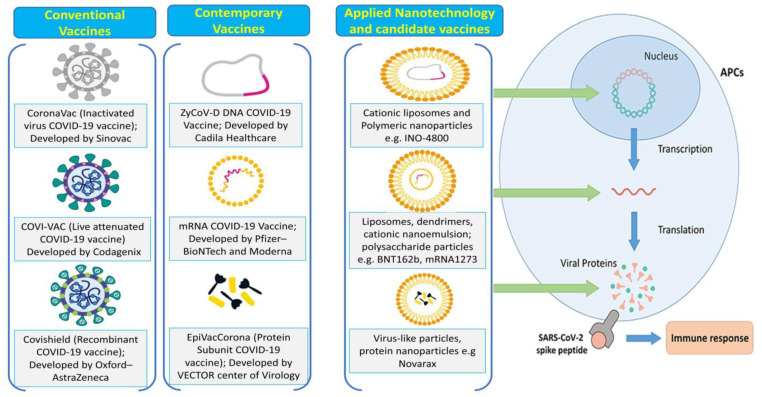
Development of conventional vaccine, contemporary vaccine, and applied nanotechnology-based vaccine against SARS-CoV-2.

**Figure 4 pharmaceutics-15-00451-f004:**
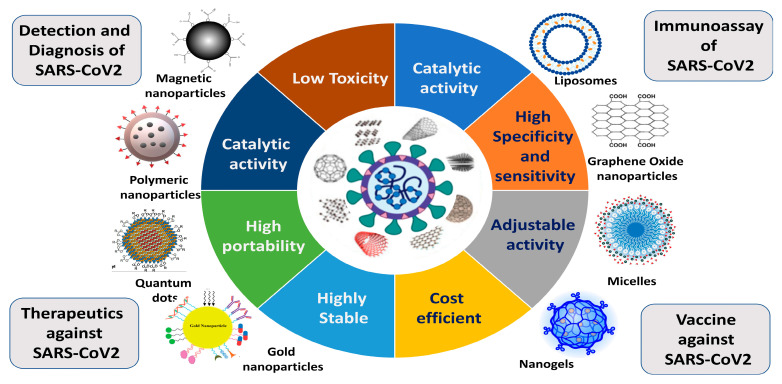
Applications of nanoparticles and the characteristics that make nanoparticles an ideal candidate.

**Figure 5 pharmaceutics-15-00451-f005:**
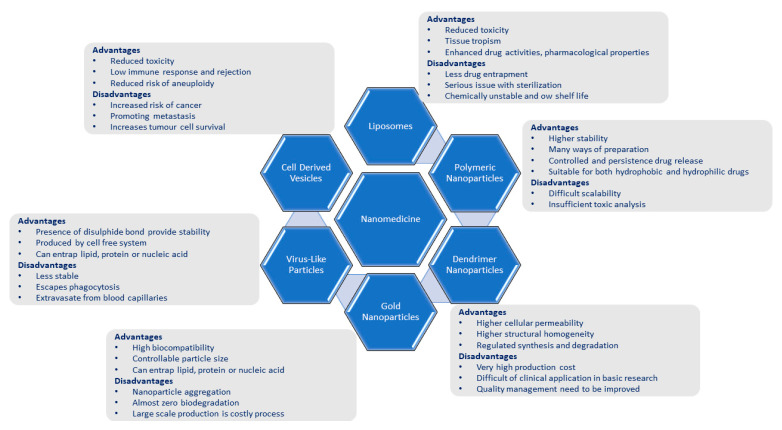
Advantages and disadvantages of the different types of nanomedicines.

**Table 1 pharmaceutics-15-00451-t001:** Merits and demerits of nanoparticle-based COVID-19 vaccines.

Presentation of Antigen	Nanoparticles as a Delivery System	Vaccine Antigens	Advantages	Disadvantages
Antigenic components packed in nanoparticle	Lipid nanoparticle/nano formulation	DNA and mRNA	Completely synthetic Increases stability and regulates transportation of nucleic acid Highly immunogenic Adjuvant not needed Customized for surface charge and lipid components	Need a specialized cold chain for storage Comparatively costly Some myocarditis-type adverse reaction occurs
	Polymer nanoparticle/nano formulation	Receptor-binding domain (RBD)	Customized for surface charge, shape, and size Targeted delivery of adjuvant and antigen Very effective for intranasal application with the use of mucosal adhesive polymer components	Low loading capability Packaging limits the immune recognition of antigenic protein and slows the immune response
Antigenic components presented in nanoparticle core	Virus-like nanoparticles/nano formulation	Spike (S) protein, membrane (M) protein, envelope (E) protein	Mimic the natural virus Rapid production can be scaled up A large number of viral proteins can be delivered	Needs complicated supportive viruses or sometimes cell lines also Restricted regulation of the viral protein structure and orientations
	Protein nanoparticles/nano formulation	Spike (S) protein and receptor-binding domain (RBD)	Increased immunogenicity compared with subunit vaccine Customized for surface charge, shape, and size At a time, together display heterologous antigenic components	No scientific data available on scalable formation Need cell line in the production and regulatory endorsement Need formulation along with adjuvant
	Micelle nanoparticles/nano formulation	Spike (S) protein	Present a large number of viral proteins in natural viral-like fashion	Very low stability No scientific data available on scalable formation
	Liposomes	Spike (S) protein and receptor-binding domain (RBD)	Completely synthetic Customized for surface charge, shape, and size At a time, together present adjuvant and antigen	Half-life and stability can be affected by the associated antigenic components No in vivo stability data available on liposome and antigenic components conjugates

**Table 2 pharmaceutics-15-00451-t002:** Nanoparticle COVID-19 vaccine candidates in clinical laboratories.

Nanoparticles Category	Developed by	Nanoparticle Vaccine Candidate [Registration Number of the Candidate]
Lipid nanoparticles	Moderna, NIAID	mRNA-1273 (LNPs) [NCT04760132]
	Pfizer/BioNTech, Fosun Pharma	BNT162b2 (3 LNP-mRNAs [NCT04760132]
	Moderna, NIAID	mRNA-1273.351 LNPs [EUCTR2021-000930-32]
	CureVac AG	CVnCoV mRNA vaccine [NCT04674189]
	Academy of Military Science (AMS), Walvax Biotechnology and Suzhou Abogen Biosciences	SARS-CoV-2 mRNA vaccine (ARCoV) [NCT04847102]
	Arcturus Therapeutics	ARCT-021 mRNA vaccine [NCT04668339; NCT04728347]
	Sanofi Pasteur and Translate Bio	MRT5500 mRNA vaccine [NCT04798027]
	Daiichi Sankyo Co., Ltd.	DS-5670a mRNA vaccine [NCT04821674]
	Elixirgen Therapeutics, Inc.	EXG-5003 [NCT04863131]
	GlaxoSmithKline	CoV2 SAM LNP [NCT04758962]
	Imperial College London	LNP-nCoVsaRNA [ISRCTN17072692]
	Providence Therapeutics	PTX-COVID19-B, mRNA vaccine [NCT04765436]
	SENAI CIMATEC	HDT-301: self-replicating mRNA-LNP vaccine [NCT04844268]
	ModernaTX, Inc.	mRNA-1283, a potentially refrigerator-stable LNP vaccine [NCT04813796]
	Chulalongkorn University	ChulaCov19 mRNA vaccine [NCT04566276]
	Shanghai East Hospital and Stemirna Therapeutics	mRNA-LNP COVID-19 vaccine [ChiCTR2100045984]
	MRC/UVRI and LSHTM Uganda Research Unit	LNP-nCoV saRNA-02 vaccine [NCT04934111]
Virus-like nanoparticles	Medicago Inc.	Coronavirus-like particle (CoVLP) [NCT04636697]
	The Scientific and Technological Research Council of Turkey	VLP vaccine [NCT04962893]
	Serum Institute of India, Accelagen Pty, SpyBiotech	RBD SARS-CoV-2 HBsAg VLP vaccine [ACTRN12620000817943]
	VBI Vaccine Inc.	VBI-2902a (Enveloped VLP of S protein) [NCT04773665]
	Radboud University	ABNCoV2 capsid VLP (cVLP) [NCT04839146]
Protein nanoparticles	SK Bioscience Co., Ltd.	RBD-I53-50 nanoparticle [NCT05007951]
	Walter Reed Army Institute of Research (WRAIR)	S protein-ferritin nanoparticle [NCT04784767]
Micelles	Novavax	SARS-CoV-2 rS/Matrix M1-Adjuvant [NCT04611802; EUCTR2020-004123-16-GB; NCT04583995]

**Table 3 pharmaceutics-15-00451-t003:** Lipid nanoparticle mRNA vaccines against coronavirus.

Vaccine Name	Encoded Antigen	Manufacturer/Developer
mRNA-1273	Spike	Moderna
BNT162b1/BNT162b2/BNT162b3	Spike	Pfizer
CureVac COVID-19 vaccine (CVnCoV)	Spike	GlaxoSmithKline/CureVac
LNP-nCoVsaRNA	Spike	Imperial College London
ARCT-021/ARCT-164/ARCT-165	Spike	Duke-NUS Medical School
ARCoV	Receptor-binding domain	AMS/Walvax/Suzhou
TAK-919	Full-length spike protein prefusion stabilized	Takeda/Moderna
ChulaCov19	Full-length spike protein	Chulalongkorn University
EG-COVID	Full-length spike protein	Eye-GENE
SW-0123	Full-length spike protein	Stemirna Therapeutics/ Shanghai East Hospital
PTXCOVID19B	Full-length spike protein	Providence Therapeutics
MRT5500 (VAW00001)	Full-length spike protein 2P modified furin cleavage site	Translate Bio/Sanofi

## Data Availability

Not applicable.

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
