# Peer review of "Aspects of Nanotechnology for COVID-19 Vaccine Development and Its Delivery Applications"

_pharmaceutics, 2023, doi:10.3390/pharmaceutics15020451_

Round 1

Reviewer 1 Report

Language errors throughout the manuscript, it needs to be corrected by a professional before publication.

Sentence in rows 25-27 needs to be revised to make it clear:

There are around 15 variants of coronavirus has been reported till now which comes from the original Wuhan variant with mutation in their genetic materials.

Rows 28-29:

It would be better to say that all the existing vaccine technologies have been exploited and after that examples can be listed.

Row 42:

m-RNA à correct to mRNA

Rows 62-63:

MERS-CoV did not cause a pandemic, this needs to be corrected.

Row 64:

Col perhaps means cold?

Rows 82-86:

Actually most commonly coronavirus (at least SARS-CoV-2) causes very mild disease, you should also bring that up. It is misleading to say that the most frequent and common symptoms of COVID-19 are high fever and so on. You should elaborate this part more.

Rows 91-94:

Revise to make clear. Also rows 94-95.

Rows 274-275 and 277-279:

The same sentence repeated twice in different rows, but different reference. This does not make sense.

Rows 375-376:

The liposomes are aspherical in shape and consist of hydrophobic components inside the sphere and hydrophobic phospholipid bilayer faces outwards [92].

è Correct hydrophobic to hydrophilic

Rows 417-418:

You cannot say that VLPs would be made of adjuvants. They are made of viruses structural proteins. If you want to say that they are made od adjuvants, you need to carefully elaborate what so you mean and put it in the context.

Rows 510-513:

Too straight generalization, you need to elaborate or correct this.

Rows 605-608:

VLPs do not have the typical activity of a virus. This is misleading, you need to correct this or very carefully elaborate what do you mean. Also rows 608-610.

Author Response

the response to reviewer 1 comments are attached

Reviewer 2 Report

This review provides aspects of nanotechnology for COVID-19 vaccine development and its delivery application. Overall it is well- organized and written. It could be considered to be accepted after addressing the following questions.

1. The authors should be aware that liposomes are different from lipid nanoparticles. A liposome is a particular form of lipid nanoparticle, it must have a lipid bi-layer structure. However, a lot of “liposome” or “liposomal” words are misused in this manuscript. Please carefully revise them.

2. Table 2, “BNT162” should be “BNT162b2”. The authors should carefully check this name throughout the manuscript.

3. The authors are recommended to highlight and introduce more of the two most popular COVID-19 vaccines, mRNA-loaded lipid nanoparticles, BNT162b2 and mRNA-1273. More discussions and a table listing all ingredients should be included.

4. The authors should discuss more challenges and perspectives regarding nanotechnology for COVID-19 vaccine development and its delivery application.

5. In conclusion, “However, COVID-19 has remained a global concern for almost a year, unlike the cases of SARS or MERS.” One year? Please revise.

Author Response

The response to reviewer 2 comments is attached.
